# Development on Health Risk Assessment Method for Multi-Media Exposure of Hazardous Chemical by Chemical Accident

**DOI:** 10.3390/ijerph17103385

**Published:** 2020-05-13

**Authors:** Hyong Jin Hong, Si Hyun Park, Hui Been Lim, Cheol Min Lee

**Affiliations:** 1Department of Nano and Biological Engineering, Seo-Kyeong University, 124, Seogyeong-ro, Seongbuk-gu, Seoul 02713, Korea; hongdonn01@skuniv.ac.kr (H.J.H.); shp8880@skuniv.ac.kr (S.H.P.); hibin016@skuniv.ac.kr (H.B.L.); 2Department of Chemical and Biological Engineering, Seo-Kyeong University, 124, Seogyeong-ro, Seongbuk-gu, Seoul 02713, Korea

**Keywords:** health risk assessment, chemical leak, multimedia exposure, excess cancer risk

## Abstract

In this study, a long-term health risk assessment was conducted for complex, multimedia exposure where the exposure duration was set for the leak of a hazardous chemical spilled after an accident. The study designed a virtual chemical accident scenario where 40 tons of benzyl chloride leaked in a factory inside the Ulsan Petrochemical Industrial Complex for one hour on 1 January 2017. Using a multimedia environmental dynamics model, benzyl chloride concentrations in air and soil were estimated. The time when the atmospheric concentration was less than or equal to the background concentration was recorded as the end point. An assessment of the cancer risk via soil ingestion was carried out after dividing the subjects into four age groups (0–9 years; 10–18 years; 19–65 years; >65 years). All age groups showed an increased cancer risk where the values exceeded 1.0 × 10^−6^. The 0–9 years age group showed the largest distribution (4.27% of the total area) with the highest maximum and mean values. The distribution maps for all age groups exhibited a trend towards the southeast of the accident site.

## 1. Introduction

The hydrofluoric acid (HF) accident in Gumi (Gumi-si, Gyeongsangbuk-do) that occurred on September 2012 led to the implementation of the Chemical Substances Control Act of 1 January 2015, in Korea. The aim of this act is to protect the public and the environment from the hazards caused by chemical substances with adequate control of such substances and ensuring rapid response to chemical accidents. This act has established institutions for the protection of workers and stable operation of relevant processes by applying a data construction method based on a number of revisions regarding the complete control of hazardous substances and the contractor reports on such substances to allow for the statistical investigation of the chemical levels [1].

According to the revised guidelines from the Statistical Investigation of Chemical Substances report by the Ministry of Environment for advanced data construction regarding the use of chemical substances, all businesses handling chemical substances are under investigation with regard to the type and amount of substances being handled as well as the control of drugs, devices and facilities that enable the response to and prevention of chemical accidents. This is to complement the basic data on the chemical accident response, prevention and workplace safety to be used by the government to minimize damage to workers who handle such substances and to contribute to the prevention of secondary damages with a reduced initial response time [2].

However, the plans have limitations to establish a system for a post-chemical accident impact assessment. Risk assessment must account for the latest environmental, ecological and topographical parameters that surpass the individual scope of conventional exposure and risk assessments for hazardous chemical substances [3]. This is due to the difficulty in applying a comprehensive and standardized method of impact assessment for chemical accidents characterized by a diversity of causal substances and different distribution characteristics of those substances, which depends on the type of accident and meteorological and topographical conditions in the accident region at the time of the accident [4].

Recently, numerous studies in Korea and overseas have suggested the development of a new model for exposure assessments that reflect multimedia and multipath exposures for a more accurate evaluation of the exposure level because exposure models have been determined to be merging various factors and variables [5,6]. For quantitative evaluations, the environmental, ecological, topographical and meteorological parameters of the accident site should be considered, and the assessment should reveal the physical properties of the chemical leaked from the accident [7].

This study aims to devise a plan for post-chemical accident impact assessments. In the study, a chemical accident scenario was simulated based on an actual chemical accident that occurred in Korea. In addition, since the hazardous chemical substance leaked through the multimedia environment during the simulated accident, the health risks to the local population caused by long-term exposure to residual substances as well as the life patterns for the development and application of an advanced method were investigated to perform health risk assessments.

## 2. Methods

### 2.1. Chemical Accident Scenario

Ulsan Metropolitan City was selected as the location of the chemical accident scenario because this city handles the largest amount of chemical substances (62,692 tons per industrial complex) [8]. The target substance, benzyl chloride, was selected among different substances defined for a response in the Chemical Substances Control Act. Among the chemical leak accidents reported by the National Institute of Chemical Safety (2015), the most serious accident was the leakage of 22 tons of hydrochloric acid from a 40-ton storage tank in a chemical factory in Ulsan in August 2015, which was caused by inadequate facility control. In reference to this case, a chemical accident scenario was designed to simulate the leakage of 40 tons of benzyl chloride in the center of the Ulsan Petrochemical Industrial Park at midnight on 1 January 2017 for a duration of one hour (Table 1).

### 2.2. Multimedia Environment Dynamics Model

The model developed in the “Development of a novel technique for post-chemical accident human impact assessment” [9], which is the precursor to this study, was used to develop the multimedia environment dynamics model to estimate the long-term multimedia behavior and extinction of a hazardous substance from a chemical leak accident. The model was developed based on a FORTRAN code to allow the formation of artificial hypotheses and analyze the behavior of hazardous pollutants according to the changes in meteorological conditions (air, soil and water systems). The modeling domain was set to a square area of 2 km^2^, with a vertical space of 10 m in the hazardous range. The accident point was set at the center with an internal grid (width = 100 m, length = 100 m) set to a nesting for a detailed concentration estimation [9].

To examine the concentrations of hazardous pollutants leaked from the chemical accident point to adjacent residential areas, the measurement scope was set to a rectangular area of 16 km × 12 km with an internal grid of 100 m × 100 m in width and length. Identical conditions were applied within a cell independent of the given topography (Figure 1). Meteorological data were recorded hourly starting from midnight on 1 January 2017, using regional weather data provided by the Korea Meteorological Administration.

### 2.3. Chronic Health Risk Assessment Following a Chemical Accident

The health risk assessment should surpass the initial risk determination based on a dose–response correlation, while comprehensively reflecting the actual human or environmental exposure-related factors by considering the amount and type of multimedia (soil, water, atmosphere) exposure, physicochemical properties of the substance and environmental dynamics such as volatilization of hazardous chemical substances, groundwater leaching, absorption into vegetation, decomposition reaction others [10]. In other words, the human exposure assessment should be a highly important component of the health risk assessment following chemical substance exposure. For the chronic risk assessment based on our scenario, the four-step health risk assessment method suggested by the U.S. Nuclear Regulatory Commission (NRC) (1993) was used [11].

#### 2.3.1. Hazard Identification

Hazard identification is a qualitative evaluation step for decide what to cause adverse health effects in humans and other animals, where the target substance data or previous studies are examined prior to risk assessment. It is a qualitative description based on the type and quality of the data, complementary information and the weight of evidence from these various sources. This step involves the screening of epidemiological data, toxicity data and in vivo and in vitro tests in toxicity databases of national and international toxicity research institutes [12].

Benzyl chloride, the target substance in this study, has been reported to cause abnormalities in the thyroid, liver and blood via various in vivo tests based on the toxicology data from international institutes, the Integrated Risk Information System (IRIS) of the U.S. Environmental Protection Agency (US EPA) and the National Institute for Occupational Safety and Health (NIOSH) (Table 2) [13,14].

#### 2.3.2. Dose–Response Assessment

The dose–response assessment is a step that numerically assesses a substance with a confirmed risk [12]. In this study, only data that contain epidemiological information, not in vivo test results, were selected. For benzyl chloride, the dose–response assessment data from the IRIS were obtained [13]. The substance was found to carry a risk of cancer upon oral administration. For benzyl chloride, dose–response assessment data from IRIS of the US EPA was used [13]. A carcinogenic slope factor (CSF) of 1.7 × 10^−1^ per mg/kg–day was used to determine the excess cancer risk (ECR) from oral exposure.

The exposure assessment involves the quantitative estimation of the human exposure level of a recipient exposed to the hazardous chemical substance. The concentration of exposure to the hazardous chemical substance is calculated by considering the exposure scenario, exposure assessment target, exposure estimation method and exposure quantification method [12].

#### 2.3.3. Exposure Assessment

The exposure assessment for the chronic health impacts of hazardous chemicals released by chemical accidents should be differentiated from a general chronic health impact assessment of hazardous chemicals. In most risk assessment studies, cumulative concentration during the exposure period is evaluated by taking the exposure duration and incorporating the estimated values (Figure 2a). However, overestimation may result in a case where the concentration shows an exponential increase immediately after the accident, followed by a rapid decline.

In this study, to reduce the uncertainties in the general method, the endpoint of the hazardous chemical substance was determined by considering the trend of the time-dependent concentration. Based on an analysis of the accurate concentration over time from the start of the chemical accident to its end point, a novel method to calculate the cumulative exposure concentration was developed (Figure 2b).

To determine the endpoint, the atmospheric background benzyl chloride concentration was examined. When the concentration of a hazardous chemical substance, leaked as a result of a chemical accident, falls below the atmospheric background concentration, the substance is considered to reach extinction. Since the background concentration of benzyl chloride is 0.01 ppb [15], the time at which the atmospheric benzyl chloride concentration declined to ≤0.01 ppb in all areas in the multimedia environment dynamics model was determined as the end point.

The concentration was calculated using the multimedia dynamics model with the determined endpoint for the chemical accident scenario, which was used to obtain the lifetime average daily dose (LADD). Here, we assume that the change in children activity patterns due to the chemical accident prevents certain activities, such as directly touching the soil, resulting in no exposure from the skin contact with soil.
(1)$$\mathrm{LADD}\;(\mathrm{mg}/\mathrm{kg}/d)=\sum_{n=1}^{CLT}\left(C_{nSoil}\times IR_{Soil}\right)/BW\left(kg\right)\times LT\left(d\right),$$

In Equation (1), CLT is the time remaining until the extinction of hazardous chemicals (d), $C_{n\;Soil}$ is the concentration of hazardous chemicals in the soil ‘n’ days after the accident (mg/kg), $IR_{Soil}$ is the soil intake rate (kg/d), BW is the body weight (in kg), and LT is the lifetime in days.

The subjects for the exposure assessment were categorized into four age groups: 0–9 years, 10–18 years, 19–65 years and >65 years. For the ingestion of benzyl chloride in the soil, the average daily dose was calculated at 80 mg/d for the 0–18 age group and 40 mg/d for the 19–65 age group [16]. The handbook published by the Ministry of Environment was used as a reference for body weight while the life expectancy from the National Key Indicators System of the Statistics Korea was used for the life span (Table 3) [17,18].

#### 2.3.4. Risk Characterization

Risk characterization was performed based on the previous risk assessment data. As benzyl chloride is a carcinogen that can be ingested through contaminated soil, a cancer risk assessment was conducted for this chemical.

Risk characterization used the slope factor of benzyl chloride obtained from the dose–response assessment as the CSF, together with the LADD from the exposure assessment, to determine the ECR based on Equation (2). The ECR is the probability that exposure to a hazardous chemical has a carcinogenic effect in one or more individuals in a given population. The World Health Organization (WHO) recommendation for the ECR for benzyl chloride is 10^−5^ (CSF from one or more individuals per 100,000) while the US EPA suggests 10^−6^–10^−4^. In this study, ECR values greater than 1.0 × 10^−6^ were used as indicators of the CSF due to benzyl chloride ingestion via soil within the region of the chemical accident.
ECR = LADD (mg/kg/d) × CSF ([mg/kg/d]^−1^).(2)

## 3. Results

### 3.1. Endpoint Determination

The endpoint of the benzyl chloride concentrations that increased as a result of the chemical accident, which began at midnight on 1 January 2017, was determined by examining the time at which the concentration was less than or equal to the standard concentration. The time-dependent distributions of the atmospheric and soil concentrations of benzyl chloride were also compared.

To visualize the results of the multimedia environment dynamics model (measurement area of 16 km × 12 km), QGIS (Ver 3.10.2; open-source geographical information system software) [19] was used to project a map with the 1:52,000 ratio scale onto the NGII (National Geographic Information Institute) topographical map. The regions in the measurement area that exceeded the atmospheric background concentration are marked in red in Figure 3. The concentrations of benzyl chloride in the air were distributed over the region from Day 1 (Figure 3a) to Day 3 (Figure 3c) and continuously decreased until Day 20 (Figure 3d). Then, the concentrations steeply decreased until it was confirmed that no region exhibited a value more than or equal to the background concentration after Day 99 (Figure 3h). The daily average concentration distributions of benzyl chloride in the air and soil within the model scope were compared (Table 4). The concentrations in the air and soil displayed a decrease in their maximum values over time. However, the minimum soil concentration increased over time, whereas no atmospheric benzyl chloride was detected after Day 3.

### 3.2. Chronic Risk Assessment

To visualize the results of the chronic risk assessment per age group via soil ingestion within the benzyl chloride chemical accident site, a QGIS [19] was used. The regions exhibiting an acceptable risk value with respect to the limit (1.0 × 10^−6^) or higher were marked in red while those with risk values ranging from 1.0 × 10^−7^ to 1.0 × 10^−6^ were marked in orange, and the remaining regions were marked in green (Figure 4).

All age groups showed an ECR zone. The lowest ECR value was calculated for the 0–9 age group, with an increasing number of ECR zones for the 10–18, 19–65 and >65 groups in that order. The ECR zone ≥1.0 × 10^−6^ was larger for the 0–9 age group while the 1.0 × 10^−7^–1.0 × 10^−6^ zone was the largest for the 10–18 group.

In the chronic health risk assessment per age group, the maximum and mean values were compared, and the regional distributions of each color zone were expressed as percentages (Table 5). Across all age groups, the maximum value was ≥1.0 × 10^−6^, indicating that the ECR and both the maximum and mean values for the 0–9 group were higher than those of the other groups. Among the ECR zones, the ≥1.0 × 10^−6^ zone showed the largest distribution for the 0–9 group while the 1.0 × 10^−7^–1.0 × 10^−6^ zone showed the largest distribution for the 10–18 group.

## 4. Discussion

Health risk assessments overcome the limitations in previous studies that lack epidemiological data by using new data based on the inferences from existing data or on novel prediction models. Thus, it is necessary to further develop new methods in the future to identify the various risk factors associated with the health impacts [20].

This study used a virtual chemical accident scenario. A large number of parameters should be considered to calculate the risks associated with exposure to hazardous chemical substances, including the extent of the leak, surrounding environment, population, the scale of the accident and the source of the accident. Thus, using all these variables to model an actual chemical accident is quite difficult. Because of the challenges in data collection, building model scenarios to enhance the capacity to make informed decisions is essential prior to setting the strategies for preparation for chemical accidents. The governments and institutions must also contribute to the implementation of these strategies through data collection and integration that can be applied to varying conditions to reduce hazards associated with chemical accidents [21].

An appropriate a multimedia environment study should reflect the toxicity in terms of path, concentration, duration and time of complex exposure (rather than simple exposure) to a hazardous chemical substance [22]. The risk assessment strategy that accounts for the exposure duration defined by the guidelines is limited by the variability of the exposure time; therefore, a risk assessment study that reflects the exposure duration based on actual cases must be developed [23]. In this study, to reduce the uncertainty in accurate exposure assessments, considered the reproducibility of the endpoint determination and different environmental media at the chemical accident site.

After the chemical accident, benzyl chloride was dispersed in the air. While the atmospheric concentration decreased, the accumulation of dispersed benzyl chloride in the soil increased with time. Although no hazard from benzyl chloride accumulation in the soil was detected in this study, the cumulative concentration in the soil was under the influence of air. The lack of background concentration data led us to use the background concentration of another media for the endpoint. Nevertheless, the alternative through a different path should reduce the ecological correlation. Setting a complementary coefficient may be necessary for studies that use different media when considering the travel path among media for hazardous chemical substances.

The results showed a zero-milligram-per-cubic-meter concentration in several regions. A chemical accident occurs in a factory or industrial complex that handles large amounts of hazardous chemicals. The surrounding areas are also potentially influenced by the accidental release of these substances. In this study, the cumulative impact of the target hazardous chemical prior to the accident was not considered; thus, we need to understand the impact(s) that may be underestimated in a simulation study.

The results of the atmospheric concentration distribution map and benzyl chloride risk assessment show that the distribution was directed toward the southeast of the accident site in all maps. The correlation between the model results and the actual meteorological environment was calculated by comparing the wind rose synoptic analysis graphs for the accident site between the onset of the chemical accident on 1 January 2017 (i.e., the time of the accident) and the endpoint on 9 April 2017 (i.e., the extinction time), using the data provided by the Korea Meteorological Administration. The comparisons revealed consistent trends (Figure 5).

## 5. Conclusions

This study analyzed a chemical accident scenario that assumed the leak of a high-concentration hazardous chemical substance to aimed to provide basic data as part of an investigation of follow-up action of the residents in the vicinity via health risk assessment of the chemical accident.

A virtual chemical accident scenario that assumed the leak of a high-volume release (40 tons) of a hazardous chemical substance was designed, and considering the end point based on exposure assessment, this study also aimed to estimate the exposure concentration of benzyl chloride.

To measure the cumulative exposure concentration over time (from start to end point), age categories were set, and the LADD was estimated using the exposure coefficient. The chronic health risk assessment per age group for soil ingestion confirmed the ECR. An ECR >1.0 × 10^−6^ was used to indicate the cancer risk by the chemical accident and revealed across all age groups. To examine the result of the health risk assessment, the distribution was graphed and the maximum and mean values were obtained to compare the distributions among the regions.

The ECR zones showed the largest distribution for the 0–9 age group, followed by the 0–18, 19–65 and >65 groups. The 0–9 age group showed the largest distribution of red zones while the 10–18 group showed the largest distribution of orange zones. Both the highest maximum and mean values occurred for the 0–9 group.

This study was conducted by performing a health risk assessment using a virtual scenario. It provides valuable information on the endpoint per hazardous chemical substance for a hazardous pollutant leaked from a chemical accident. The results provide a foundation for the development of future chronic health risk assessments.

## Figures and Tables

**Figure 1 ijerph-17-03385-f001:**
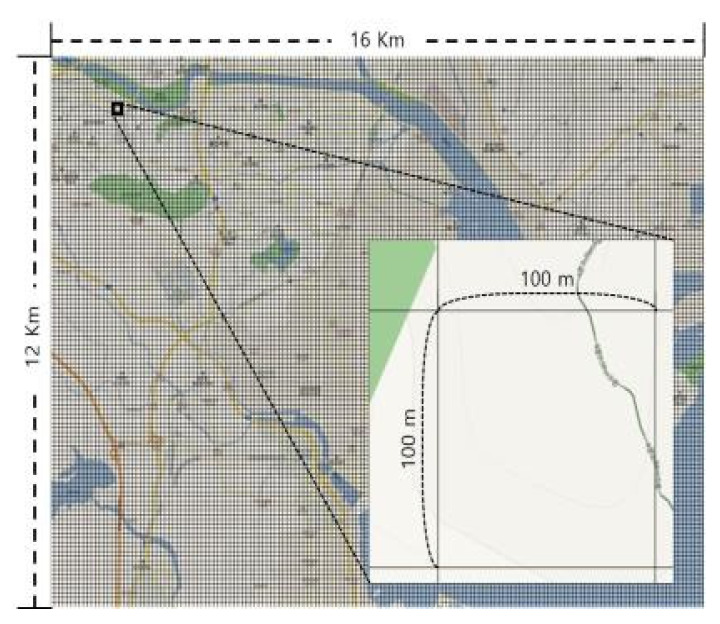
Operating range of the multimedia environmental monitoring model.

**Figure 2 ijerph-17-03385-f002:**
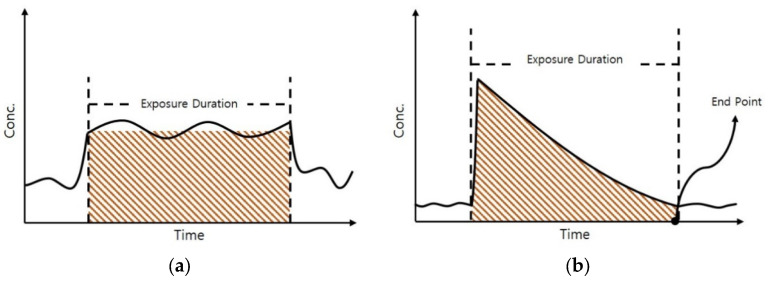
Exposure assessment method: (**a**) cumulative concentration, and (**b**) cumulative concentration from chemical accidents.

**Figure 3 ijerph-17-03385-f003:**
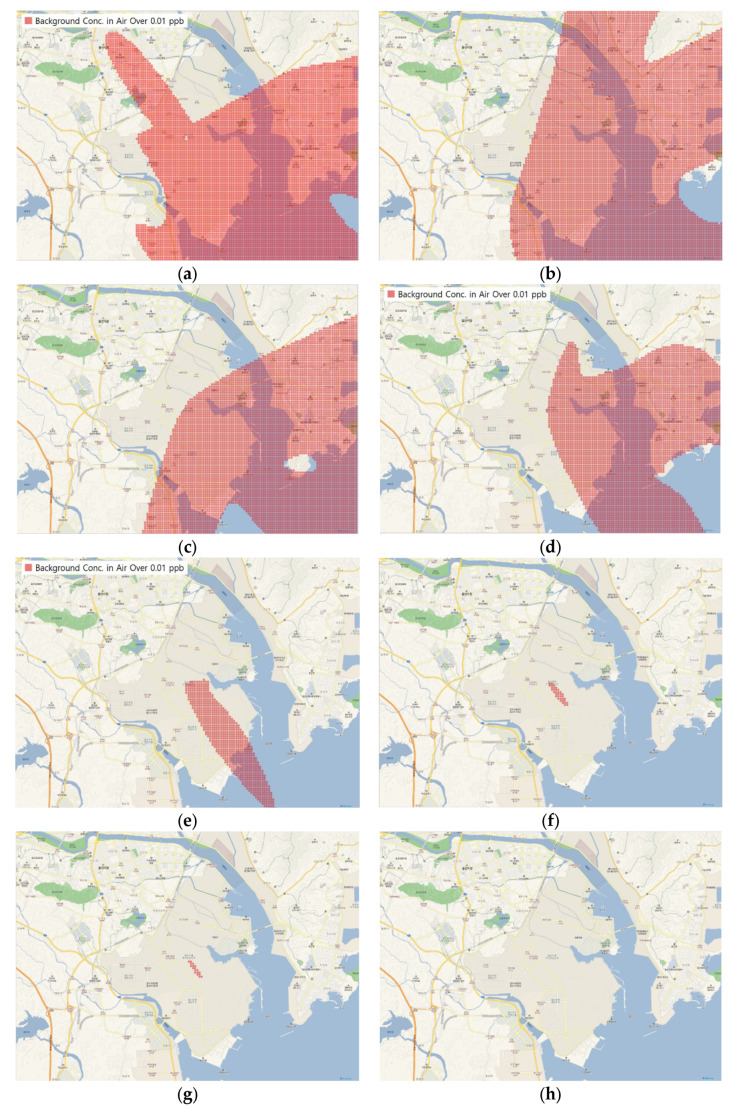
Concentration distribution maps of benzyl chloride in the air: (**a**) Day 1 (1 January 2017); (**b**) Day 2 (2 January 2017); (**c**) Day 3 (3 January 2017); (**d**) Day 20 (20 January 2017); (**e**) Day 40 (9 February 2017); (**f**) Day 80 (20 March 2017); (**g**) Day 97 (6 April 2017); (**h**) Day 99 (8 April 2017).

**Figure 4 ijerph-17-03385-f004:**
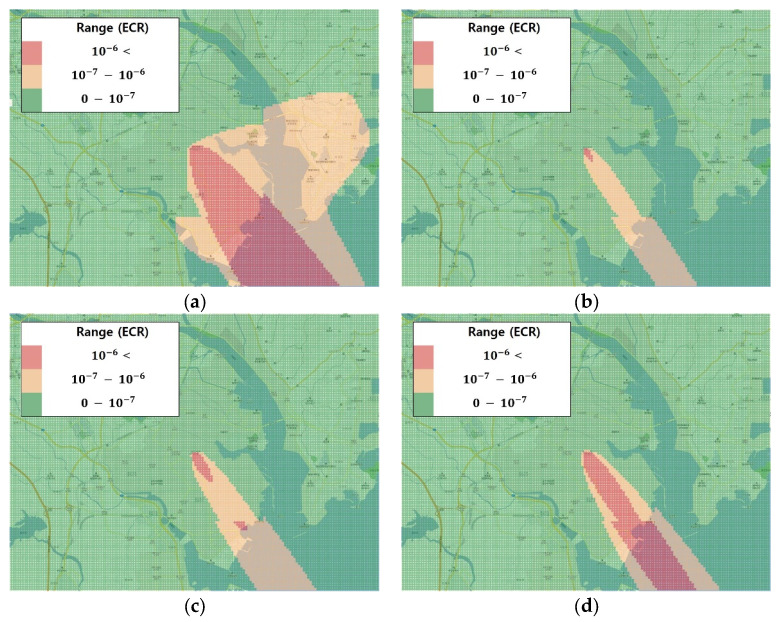
Long-term health risk assessment by age group. ECR: excess cancer risk. (**a**) 0–9 years of age; (**b**) 10–18 years of age; (**c**) 19–65 years of age; (**d**) 65 years of age and older.

**Figure 5 ijerph-17-03385-f005:**
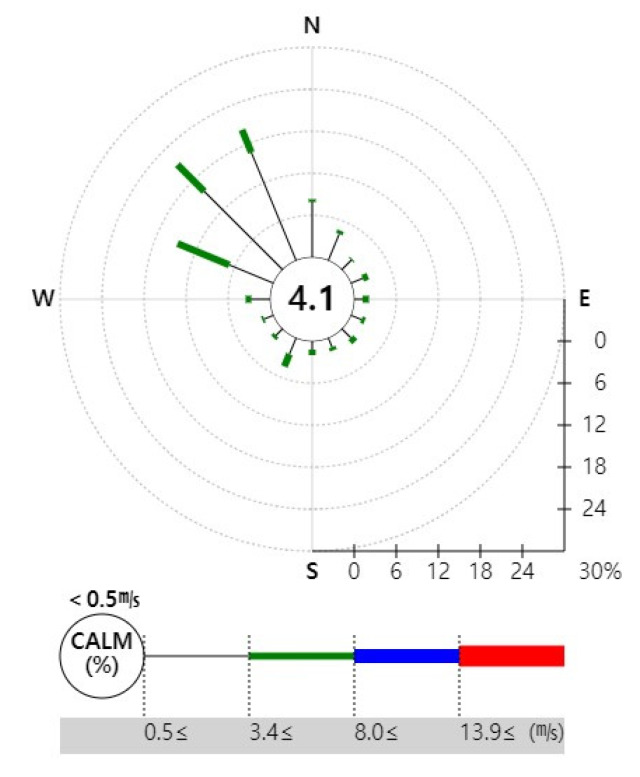
Wind rose of chemical accident period (1 January 2017–8 April 2017).

**Table 1 ijerph-17-03385-t001:** Chemical accident scenario conditions.

Parameters	Value
Accident date	01/01/2017 00:00
Accident point	Ulsan Petrochemical Industrial Complex
Type	Leak
Pollutant	Benzyl chloride
Amount	40 tons
Time	1 h

**Table 2 ijerph-17-03385-t002:** Acute toxicity studies of benzyl chloride.

Physical State	Method	Species	Toxicity (Value)	Result	Reference
Liquid	Linearized multistage procedure, extra risk	Rat (F-344)	Slope factor (0.71 mg/kg)	Thyroid, C-cell adenoma/carcinoma	IRIS EPA US.
Liquid	Linearized multistage procedure, extra risk	Rat (F-344)	Unit risk (0.49 µg/L)	Thyroid, C-cell adenoma/carcinoma	IRIS EPA US.
Liquid	Linearized multistage procedure, extra risk	Rat (F-344)	(1 in 10,000) (0.2 µg/L)	Thyroid, C-cell adenoma/carcinoma	IRIS EPA US.
Liquid	Linearized multistage procedure, extra risk	Rat (F-344)	(1 in 100,000) (2 µg/L)	Thyroid, C-cell adenoma/carcinoma	IRIS EPA US.
Liquid	Linearized multistage procedure, extra risk	Rat (F-344)	(1 in 10,000) (0.2 µg/L)	Thyroid, C-cell adenoma/carcinoma	IRIS EPA US.
Liquid	Acute toxicity test	Mouse	LD50 (1500 mg/kg)		NIOSH
Liquid	Acute toxicity test	Rat	Lowest published toxic dose (100 mg/kg)	Liver: other changes, Blood: other changes, Biochemical: enzyme inhibition, induction or change in blood or tissue levels: other transferases	NIOSH

**Table 3 ijerph-17-03385-t003:** Exposure factors.

Group (Age)	Soil Intake (mg/d)	Body Weight (kg)	Lifetime (d)
0–9	80	13.3	30,186
10–18	80	53.6	
19–65	40	63.3	
>65	40	60.7	

**Table 4 ijerph-17-03385-t004:** Exposure factor.

Group (age)	Maximum	Average	Red ^1^ (%)	Orange ^2^ (%)	Green ^3^ (%)
0–9	3.57 × 10^−5^	1.05 × 10^−7^	4.27	3.54	92.19
10–18	8.86 × 10^−6^	2.61 × 10^−8^	0.37	5.88	94.12
19–65	3.75 × 10^−6^	1.11 × 10^−8^	0.07	4.30	95.63
>65	3.91 × 10^−6^	1.15 × 10^−8^	0.05	4.02	95.93

^1^ ECR > 1.0 × 10^−6^, ^2^ ECR = 1.0 × 10^−7^–1.0 × 10^−6^, ^3^ ECR < 1.0 × 10^−7^.

**Table 5 ijerph-17-03385-t005:** Distribution of risk assessment.

Type	Date (day)	Air		Soil	
		Maximum (mg/m^3^)	Minimum (mg/m^3^)	Maximum (mg/kg)	Minimum (mg/kg)
Daily	1	43.10	6.43 × 10^−13^	90.51	1.93 × 10^−13^
	2	9.54	6.00 × 10^−16^	62.29	4.00 × 10^−13^
	3	8.17 × 10^−8^	0	35.33	6.04 × 10^−12^
	10	2.55 × 10^−3^	0	12.60	7.36 × 10^−7^
	40	3.83 × 10^−4^	0	4.24	3.21 × 10^−6^
	80	7.50 × 10^−5^	0	1.41	2.99 × 10^−6^
	97	4.94 × 10^−5^	0	1.15	2.74 × 10^−6^
	99	2.23 × 10^−12^	0	1.12	2.68 × 10^−6^

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
