# Peer review of "Development on Health Risk Assessment Method for Multi-Media Exposure of Hazardous Chemical by Chemical Accident"

_ijerph, 2020, doi:10.3390/ijerph17103385_

Round 1
Reviewer 1 Report
IN GENERAL
In this paper, a health risk assessment method for multimedia exposure to chemical accidents of hazardous chemicals is established based on a specific hypothetical accident, and I think both the evaluation method and the evaluation result are credible. However, this paper looks more like a risk assessment report than a research paper. The author should focus on the analysis of the shortcomings of the current risk assessment method, how to improve the risk assessment method, and this paper should verify the application according to the improved assessment method.
IN PARTICULAR
1.In the abstract, the authors should introduce the necessity of this research and the value of the main results in order to better promote this article.
- Introduction should be rewritten. What are the new requirements from Chemical Substances Control Act and the revised guidelines from the Statistical Investigation of Chemical Substances, and What are the problems with the existing evaluation methods and evaluation results in implementing these requirements.
- In the Methods, how to improve and improve the existing evaluation methods is the focus. This paper should simplify the introduction of existing methods and referenced methods, can summarize the existing methods by listing them.
Author Response
[2020-04-27]
Dear Reviewer
I wish to submit a research article for publication in International Journal of Environmental Research and Public Health, titled “Development on health risk assessment method for multi-media exposure of hazardous chemical by chemical accident .” The paper was coauthored by Hyong Jin Hong, Si Hyun Park, and Hui Been Lim.
In this study, we conducted a long-term health risk assessment for complex, multimedia exposure where the exposure duration was set for the leak of a hazardous chemical spilled after a chemical accident. We designed a virtual chemical accident scenario for benzyl chloride leakage in a factory inside the Ulsan Petrochemical Industrial Complex. Using a multimedia environment dynamics model, we estimated benzyl chloride concentrations in air and soil, and recorded the end points. We assessed the cancer risk via soil ingestion after dividing the subjects into four age groups. All age groups showed an increased cancer risk where the values exceeded the standard limit. The 0–9 years age group showed the largest distribution with the highest maximum and mean values. We believe that our study makes a significant contribution to the literature because it provides valuable information on the endpoint per media for a hazardous pollutant leaked from a chemical accident. The results provide a foundation for the development of future chronic health risk assessments.
Further, we believe that this paper will be of interest to the readership of your journal because International Journal of Environmental Research and Public Health publishes research in environmental health sciences and public health and focuses on the impacts of natural phenomena and anthropogenic factors on the environmental quality, which overlaps with the objectives of our study.
Thank you for your consideration. I look forward to hearing from you.
Sincerely,
Cheol Min Lee
Department of Nano and Biological Engineering
Seo-Kyeong University, 124
Seogyeong-ro, Seongbuk-gu, Seoul, Republic of Korea
Email: cheolmin@skuniv.ac.kr
010-5276-8040
02-940-7616

Reviewer 2 Report
This manuscript describes the use of a multi-media environment dynamics model to estimate the behavior of an accidental hazardous chemical release, and predict levels of contaminant over time and use this long-term model to predict the Excess Cancer Risk (ECR) of a potentially exposed population.
The authors applied the multi-media environment dynamics model to estimate the long-term multi-media behavior and
extinction of a hazardous substance caused by an accidental release of 40 tons benzyl chloride. The authors applied these data to estimate ECR in an age stratified local population. The authors found that there were exposure zones where the ECR may be greater than 1:1,000,000 across all age groups but, the youngest group (0-9 years) had the most widely distributed ECRs and the highest maximum and mean Exposure Factors.
The manuscript would be improved with careful review for English and style.
Introduction
The Introduction is concise but, is not completely clear in defining the purpose of the study. There is information in the Materials and Methods that would be better included in the introduction. Typically, the first paragraph in sub-section would be more appropriate in the introduction or discussion.
Materials and Methods
Section 2.1 lines 76-78, this sentence "Among the leak accidents reported by the National Institute of Chemical Safety (2015), the most serious leak accident was the leakage of 22 tons of hydrochloric acid from a 40-ton storage tank in a chemical factory in the Ulsan region in August 2015, which was caused by inadequate facility control" is not relevant to the materials and methods.
Section 2.2. While the model is referenced (Lee 2019) a short description of how the model operates would be warranted, perhaps a simple summary can be provided.
Section 2.3 Lines 103-108, "The health risk assessment method should surpass the initial risk determination, such as acute and aquatic toxicity, based on a dose-response correlation while comprehensively reflecting the actual human or environmental exposure-related factors by considering the amount and type of use, physicochemical properties, and environment dynamics [10]. In other words, the human exposure assessment should be a highly important component of the health risk assessment following chemical substance exposure." should be removed. However, the authors should summarize the US NRC 4-Step Health Risk Assessment method used in the current study.
Section 2.3.1 This reviewer recommends the authors not use scientific notation in Table 2. for example use 0.71 mg/kg instead of 1.7E–1 mg/kg. Also there is no need to include both Risk Level(E-4) and 1 in 10,000. 1 in 10,000 is sufficient.
Section 2.3.2 Table 3 is not needed. Simply state, "For benzyl chloride, dose-response assessment data from IRIS of the US EPA was used [13]. A carcinogenic slope factor (CSF) of 1.7 x 10 -1 per mg/kg-day was used to determine the excess cancer risk (ECR) from oral exposure."
Section 2.3.3 Lines 134 through 148 including Figure 2 are more appropriate for the introduction.
Results
Section 3.1 Caption of Figure 3. It would be more clear to describe the figures as representing days since the incident (Jan 1). This reviewer suggests: Concentration distribution maps of benzyl chloride in the air: (a) Day +1; (b) Day +2; (c) Day +3; d) Day +20; (e) Day +40; (f) Day +80; (g) Day +97; (h) Day +99
Section 3.2 Figure 4. The legend should be able to describe the figure. Also, the age groups on the figure do not match the narrative in lines 220-223.
Discussion
The discussion could be better organized. The authors should describe findings of note for the current study results and discuss or interpret those in the context as to why these results are novel.
Conclusion
The conclusion should be a concise summation of the study and key findings. This is not clear as written.
Author Response

(The authors gave the same response as above.)

Reviewer 3 Report
See attached PDF.

Author Response

(The authors gave the same response as above.)

Reviewer 4 Report
Line 32: The phrase “the aim of this act is to prevent public health…” needs revision.
Authors mention model limitations, uncertainties and the issue that has to be dealt regarding the fluctuating concentrations of the chemicals in the environment. Furthermore, it was chosen that human exposure and risk assessment would be investigated for only one toxic substance although it is widely known that in such accidents as described in the study, various toxic chemical substances are emitted and what is more, environmental reactions can take place resulting in even more toxic byproducts. It is mentioned neither in the introduction nor in discussion the increased hazard and public health risk due to the exposure to more than one toxic substance that should have been pointed out. Authors should present the complexity of this issue, mention the limitations of the model that was used, future perspectives and discuss/present recent toxicological studies (doi: 10.1016/j.toxlet.2019.04.005. ;) and risk assessment models that have been developed (doi: 10.1016/j.toxrep.2019.06.010. ) and they have taken into consideration the cumulative long term exposure at low concentrations of various pollutants (DOI: 10.1016/j.toxrep.2018.10.010 ; ) and calculation of the hazard index even for compounds that have different adverse effects.
Author Response

(The authors gave the same response as above.)

Round 2
Reviewer 1 Report
My comments have been well implemented and I have no new comments. However, the author may have neglected to provide a point-to-point response file to the review comments.
Author Response
Pretty thanks to make up for my fault.
I did to add my article with your advice.
Your advice helps to modify my article that some misunderstanding things.
I humbly admitted my lacking study.
Thanks to your advice again.
Please, take care yourself for viruses
Nowadays could be serious.

Reviewer 2 Report
The authors were not completely responsive to review comments. The manuscript is improved but, reads more as an application of a previous developed model to predict exposures and perform a risk assessment for a specific chemical and location.
Some specific comments follow:
page 3 line 84: The accident point was set at the center with an internal grid set to a nesting of 100 m2 (width = 100 m, length = 100 m) for a detailed concentration estimation [9]. 100m x 100m is 10,000 m2.
page 3 line 98- define physiochemical properties incorporated into the model (solubility, vapor pressure, boiling point, vapor density etc)
page 4 line 108-112 and Table 2. Table 2 is not needed. Only EPA IRIS data for cancer endpoint was used.
page 6 line 181-183. Better to describe QGIS as a free, open source Geographic Information System and provide the web link for information and download.
page 8, line 204 "a geographical information system (GIS) was used" this GIS needs to be defined/described
page 9, lines 225-228. This is confusing as written. Clarify or delete. The Discussion and Conclusions are still very long and not focused. The Discussion should describe the research i.e. test of a model of multi-media exposure routes based on a dynamic exposure/risk assessment model. The manuscript does not really describe the development of the model but evaluates the results of a simulated spill and uses real weather, geographic, physiochemical characteristics, and toxicological information to develop an improved risk assessment based on an improved dynamic exposure model.
the Conclusion may be summarized by:
This study analyzed a chemical accident scenario that assumed the leak of a high-volume release (40 tons) of a.hazardous chemical substance to investigate the time-dependent mobile dynamics among multiple media and environments. Considering the end point based on the media, this study also aimed to provide basic data as part of an investigation of post-accident measures via a health risk assessment of the residents in the vicinity of the chemical accident point.
The chronic health risk assessment per age group for soil ingestion revealed the ECR across all age groups. In this study, an ECR >1.0 × 10–6 was used to indicate the cancer risk due to soil ingestion for regions affected by the chemical accident. To examine the result of the health risk assessment, the distribution was schematized and the maximum and mean values were obtained to compare the distributions among the regions. The regions exhibiting an acceptable risk value with respect to the limit (1.0 × 10–6) or higher were marked in red while those with risk values ranging from 1.0 × 10–7 to 1.0 × 10 –6 were marked in orange, and the remaining regions were marked in green The ECR zones showed the largest distribution for the 0–9 age group, followed by the 0–18, 19–65, and >65 groups. The 0–9 age group showed the largest distribution of red zones while the 10–18 group showed the largest distribution of orange zones.
Author Response
Dear reveiwer
Pretty thanks to make up for my fault.
I did to add a article with your advice.
I have to explain about 'Hazard identification' at page 3 line 104-115.
Hazard identification examines the capacity of an agent to cause adverse health effects in humans and other animals (US EPA 1995 a). It is a qualitative description based on the type and quality of the data, complementary information (e.g. structure–activity analysis, genetic toxicity, pharmacokinetic) and the weight of evidence from these various sources. These why I have to use the table. It is one of step for Risk assessment. So I removed and modified some things.
Based on your advice, I modified Page 3, 6, 8, 9.
Your advice helps to modify my article that some misunderstanding things.
I humbly admitted my lacking study.
Please, take care yourself for viruses
Nowadays could be serious.
Thanks to your advice again.
